# Decentral Hydrogen

**Paul Grunow**

Energy Systems Group, Trinity Solarbeteiligungen GmbH, Marienburger Str. 31a, 10405 Berlin, Germany; p@ulgrunow.de; Tel.: +49-173-614-3992

**Abstract:** This concept study extends the power-to-gas approach to small combined heat and power devices in buildings that alternately operate fuel cells and electrolysis. While the heat is used to replace existing fossil heaters on-site, the power is either fed into the grid or consumed via heat-coupled electrolysis to balance the grid power at the nearest grid node. In detail, the power demand of Germany is simulated as a snapshot for 2030 with 100% renewable sourcing. The standard load profile is supplemented with additional loads from 100% electric heat pumps, 100% electric cars, and a fully electrified industry. The renewable power is then scaled up to match this demand with historic hourly yield data from 2018/2019. An optimal mix of photovoltaics, wind, biomass and hydropower is calculated in respect to estimated costs in 2030. Hydrogen has recently entered a large number of national energy roadmaps worldwide. However, most of them address the demands of heavy industry and heavy transport, which are more difficult to electrify. Hydrogen is understood to be a substitute for fossil fuels, which would be continuously imported from non-industrialized countries. This paper focuses on hydrogen as a storage technology in an all-electric system. The target is to model the most cost-effective end-to-end use of local renewable energies, including excess hydrogen for the industry. The on-site heat coupling will be the principal argument for decentralisation. Essentially, it flattens the future peak from massive usage of electric heat pumps during cold periods. However, transition speed will either push the industry or the prosumer approach in front. Batteries are tried out as supplementary components for short-term storage, due to their higher round trip efficiencies. Switching the gas net to hydrogen is considered as an alternative to overcome the slow power grid expansions. Further decentral measures are examined in respect to system costs.

**Keywords:** hydrogen storage; prosumers; combined heat and power devices; flexibility options

## 1. Introduction

Does it make sense to use the conversion heat from hydrogen storage for low-temperature heating in buildings? This question has not yet been investigated in terms of cost effectiveness on the system level in an all-electric energy system.

In detail, the study models the benefits of combined heat and power plants based on integrated fuel and electrolysis cells, named *Storage CHP* [1]. They are introduced into a simulation of a 100% renewable and sector-coupled energy system in terms of scalable heating devices in buildings. On the one hand, they increase the system's efficiency through local heat-coupling, while, on the other hand, they are able to balance the power grid through several million decentral devices node by node. The resulting network structure is more robust and redundant, as proposed in "The Cellular Approach" [2,3]. It addresses the following essentials:

1.  *Fluctuation balancing* in a 100% renewable energy system with intermittent photovoltaic and wind as its two main pillars, and limited continuous power contributions from biomass and hydropower [4]
2.  *Peak shaving* of the power demand from heat pumps on cold winter days at low power generation ("Kalte Dunkelflaute")

3. *Demand-side management* through *Storage CHP*, which switches from power consumption to generation and vice versa, as triggered by the voltage and frequency of their nearest grid nodes, while building heating is the primary use on-site. Less *grid expansion* is needed while hydrogen is stored and transported in a repurposed gas grid (or initially local tanks), even if electricity demand multiplies in 100% sector coupling.

Why decentral hydrogen? This study will conclude that fuel cells are favorable over gas turbines and batteries, in contrast to other studies, e.g., on Germany [5] or Europe [6,7], or indeed on a global scale [8]. However, these studies provide a good overview about the actual status, and are used as reference here, including their technical reference lists. The investment and operating costs are taken from [8], and are listed in Table A1 in the Appendix A.

In this study, the distributed heat-coupling of fuel and electrolysis cells is exploited to reduce the power and number of heat pumps, and therefore the overall power demand and costs [9]. Hydrogen is chosen over synthetic fuels, due to its higher round-trip-efficiency. Finally, fuel cell and electrolysis stacks are freely down scalable to local needs, while their efficiencies do not scale with stack size. In this aspect, surface conversion processes differ from bulk conversion processes, e.g., batteries, fuel, and electrolysis and photovoltaic cells, in contrast to gas turbines [10]. The actual gas route appears attractive due to existing infrastructures and supply chains, but not because of costs, as will be demonstrated out below. Furthermore, huge invests from independent countries following their own roadmaps are required, which bears new unpredictable dependencies.

District heating is an obvious option in cities. However, due to their high transport losses of >10% and thus higher operating temperatures of >80 °C, the efficiency of central heating pumps is principally lower than that the same pump on-site. Although district heating offers an uncomplicated alternative to replace fossils, the heat will not be an inexpensive by-product any longer. It is shown that scalable decentralized hydrogen plants are much better suited to decentral power generation by wind and photovoltaics than any central power or heat plant.

The LUT team places batteries on the Mediterranean borders [6] to power northern countries via pan-European power grids, i.e., they make continental infrastructures a prerequisite. However, it should be noted that only 15% of the German power grid expansion plans from 2011 could be realized by the end of 2021. These cover 1700 km out of 12,250 km [11], including the critical north–south power connection addressed again in Section 3

Where these plans are based on Germany's actual power needs, i.e., before doubling it in sector coupling, see Appendix A Table A2 for more information.

Although gas turbines, batteries, and grid expansions are essential pillars in most actual master plans, some groups have successfully modelled hydrogen fuel cells as a cost-effective alternative, e.g., for Switzerland [12], while batteries remain favorable in southern Europe, according to the same authors. The role of decentral hydrogen is discussed for self-sufficient homes [13,14], small hydrogen networks [15–17], and in respect to new market designs with grid penalty costs [18] or time-dependent feed-in tariffs [19].

## 2. Storage CHP: A Complement to Wind and Photovoltaics

The *Storage CHP* concept includes a small fuel cell CHP with an additional electrolysis in thermal coupling. It enables power-to-hydrogen conversion and its re-conversion as an integral part of private homes or commercial real estate (see company HPS) [20]; Figure 1 shows an integrated CHP system indoors and hydrogen bundles outdoors. Its bidirectional functionality will be used in the following simulation to balance the power grid on a larger scale, whereby the automatic switch-over is triggered by the grid frequency or grid voltage at the specific grid node. Conventional inverters already have to operate this way, due to existing German grid codes. The heat is used entirely for on-site building heating and warm water supply in Figure 2, replacing all fossil heaters in the frame of this simulation.

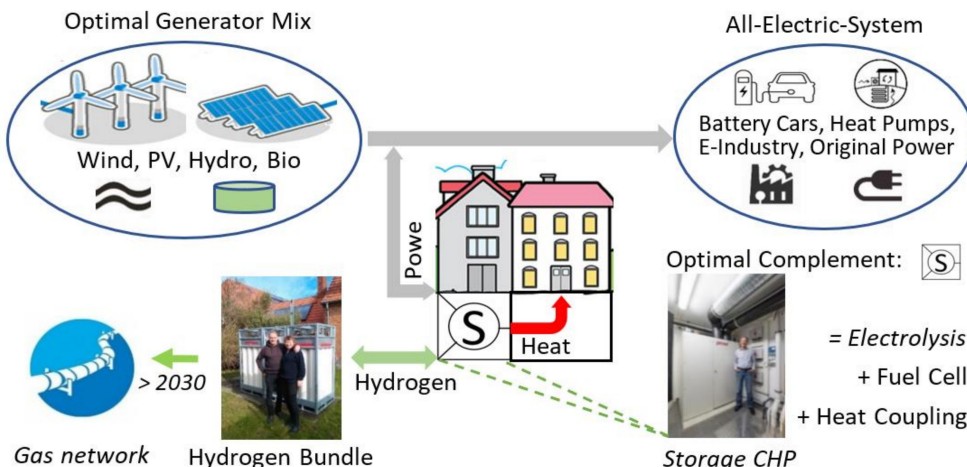

**Figure 1.** Decentral 100% renewable energy system with *Storage CHP*.

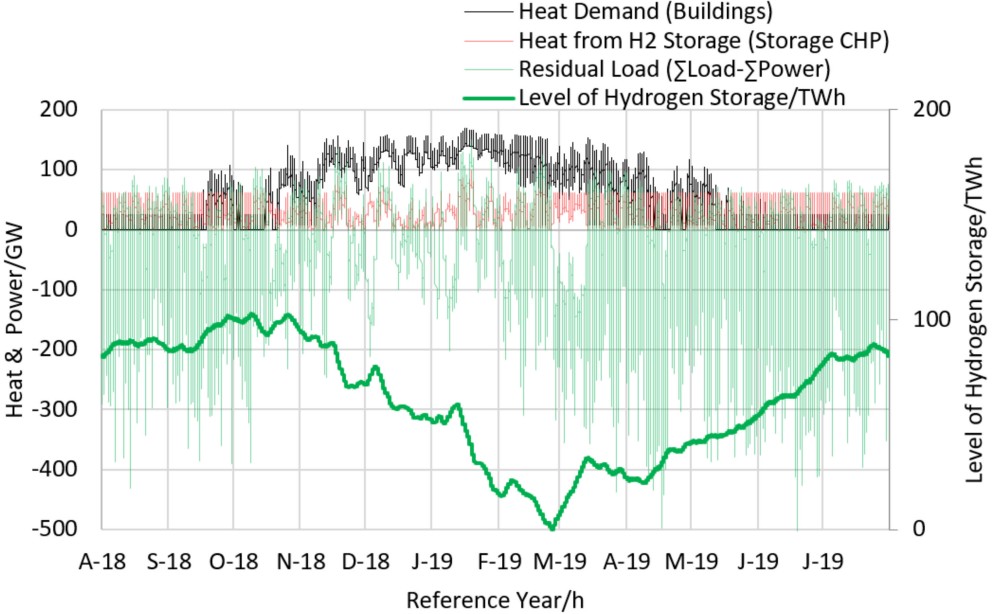

**Figure 2.** Heat from hydrogen conversion in a building integrated with *Storage CHP* covers significant shares of the heat demand in Germany's buildings in 2030, especially in winter (center). Negative residual power (green) is used for electrolysis, positive residual power starts up the fuel cells. Curtailment: 3.4%. Hydrogen storage capacity is 103 TWh (right axis) for the standard scenario with 400 GW wind.

Figure 3 demonstrates that any mix of wind power and photovoltaics can be balanced completely by *Storage CHP*. This allows independent energy cells to form, even without intercellular power exchange, although at different system costs (see Figure 3). This would offer a higher degree of redundancy, robustness, and, finally, reliability to the power grid. It also enables faster roll outs: investments therein are executed by prosumers through the replacement of their heating system instead of slow developments in new infrastructure, such as power grid expansions and central hydrogen facilities. The latter also offers no or less efficient thermal coupling, as is the main topic of discussion here.

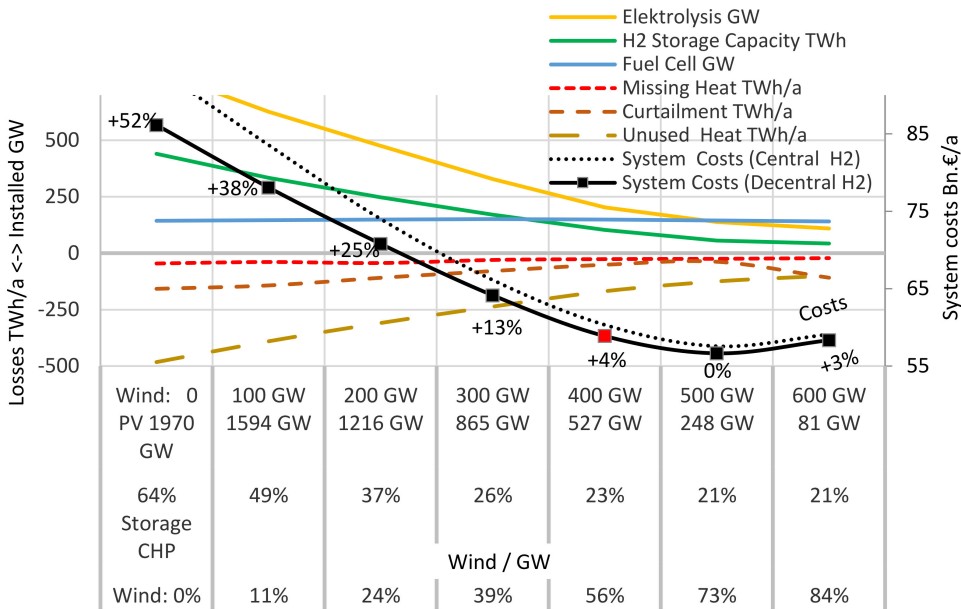

**Figure 3.** System costs (right axis) with and without *Storage CHP* (decentral) in a 100% renewable all-electric Germany for various wind shares. The red dot marks the standard scenario, with 400 GW wind used in the following scenario.

The simulations are based on hourly yield data from ISE Fraunhofer energy charts and Deutscher Wetterdienst (DWD) ambient temperature data for Frankfurt am Main between August 2018 and July 2019 ranging from −2.7 °C to 42.5 °C. The latter determines the electrical load of the heat pumps, as well as the corresponding heat demand of the respective building (see Appendix A).

An all-electric Germany of 2030 is modeled with 100% electric cars, fully electrified industry, and either *Storage CHP* or electric heat pumps for building heating. At present, 21 million heating systems are in operation in Germany, with around 50% connected to the gas network. The existing German gas network has a storage capacity of 300 TWh, which is a limiting parameter [10]. The usable capacity is only 100 TWh for hydrogen due to its calorific value being three times lower than natural gas.

With respect to this, local hydrogen tanks offer a practical alternative, especially for getting started, as long as regulations for hydrogen injection into the gas net are not in place. According to Yang and Ogden [21], hydrogen bundles or tanks on trucks are as cost-effective as hydrogen pipelines over short distances (<50 km) and volumes (<15 t/day). On the other hand, the use of existing gas pipelines equipped with appropriate compressors would reduce the costs by a factor of five over new pipelines [22]. The gas network might be used for hydrogen storage and distribution after a possible gas exit (which has become very urgent in the current geo-political situation of 2022). Photovoltaic, wind, fuel cell, electrolysis power, and battery capacities are set as free parameters, while hydropower and biomass are kept fixed at 7.76 GW and 20 GW, respectively, with flat generation profiles and a summed cost of EUR 6.05 bn/annum from [8]. The share of *Storage CHP* over heat pumps was categorized as an additional free parameter. Further limitations and assumptions are explained in the Appendix A.

A cost minimum is achieved with 494 GW for wind and 261 GW for photovoltaic capacity, and a share of 20.7% for storage CHP, as compared to 89.3% for heat pumps. Fuel cell and electrolysis sum up to 146 GW, and 135 GW is distributed over four million buildings with *Storage CHP* versus 17 million with heat pumps. The system costs amount to EUR 56.6 billion/annum, excluding transport and electrified loads, i.e., electric cars, electric heat pumps, and an electric industry. For comparison, fossil fuel imports to Germany cost EUR 64.9 billion/annum and EUR 60.8 billion /annum in 2018 and 2019, respectively [23], i.e., an optimal renewable energy system would already save around 10% in annual costs.

The cost minimum results from decreasing electrolysis power, curtailment, missing heat, and unused (or lost) heat when the wind power capacity is increased is shown in Figure 3. *Missing heat*, as well as *lost heat*, is due to temporary mismatches between local heat demand and residual power at the respective grid node. This is because the *Storage CHPs* are operated/power-controlled by the grid frequency and voltage via the feed-in inverters (or, alternatively, by software, as preferred by the power utilities), but not heat-controlled through the on-site thermal demand. In total, this amounts to 12% of the available power, which is still not usable as *lost heat*, as shown in Figure 4.

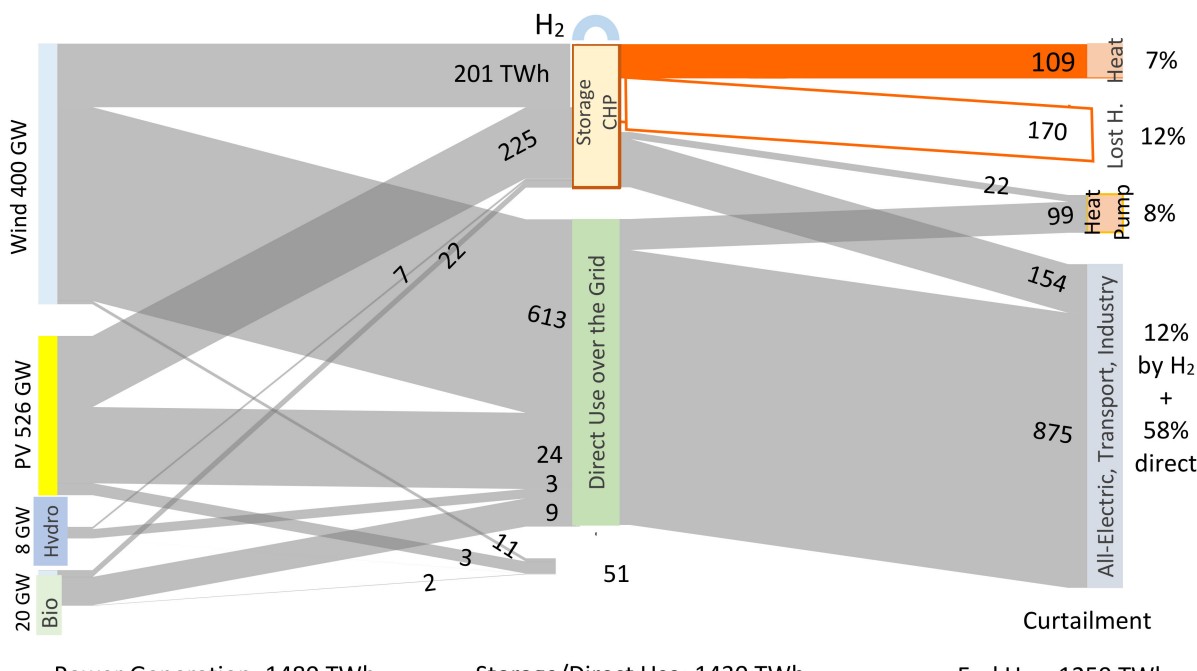

**Figure 4.** Energy flows for an all-electric renewable Germany as a snapshot of 2030. Decentral *Storage CHP* compensates for 100% of the power fluctuations and provides heat (red) for 23% of the buildings, while 77% are heated by heat pumps. Remark: The heat pumps achieve a much higher heat coverage, since their input power is multiplied by their coefficient of performance COP = 3.2.

The necessary fuel cell power is calculated directly from the maximum residual load on 24 January 2019 at 7 p.m. Without thermal coupling, the calculated total fuel cell power meets the central scenario with 160 GW for gas turbines in [5]. This confirms a good agreement in boundary conditions and assumptions (see Appendix A), and demonstrates that 2018/2019 was a good reference winter.

According to [5], onshore wind is limited to 230 GW (3.1% of Germany's area for 21 MW/km$^2$) and offshore wind to 80 GW. Since the latter performs with a higher yield, i.e., a factor of 2, an effective onshore value of 400 GW is used as a standard scenario in Figure 3, as well as in the following text. Within this wind limit, the present algorithm is calculated to be 526 GW for photovoltaics, which is close to the 530 GW calculated in [5].

In contrast to [5] and [6], batteries reveal no benefit in the present cost simulation. In an extended sensitivity analysis, the investment and operating costs of the batteries were lowered by around 80% (e.g., to EUR 30/kWh for CAPEX). This creates a new cost minimum for an added battery storage capacity of around 500 GWh (for energy-to-power ratios varied between 1 and 6), which is also in good agreement with [5]. However, this capacity is far too low to cover the calculated 27 TWh gap for long-term storage, but would reduce costs for photovoltaics (−2%) and electrolysis (−3%) power. Due to the large gap, the fuel cell costs affect the resulting power mix and, subsequently, total costs. For example,

if the fuel cell's CAPEX increases by a factor of 10, the system's power cost per kWh would increase from EUR 0.052/kWh to EUR 0.072/kWh (+38%). The 100% renewable system would then cost 15% more than the gas-coupled system in [5], at EUR 0.063/kWh (see Appendix A Table A3). On the other hand, the fuel cell costs in Table A1 correspond to the introductory prices of the two actual hydrogen car suppliers, who sell their cars at EUR 40,000 above the market price for internal combustion engine cars. With a fuel cell output of 100 kW per car, the fuel cell's target costs would be below EUR 400/kW, compared to the EUR 471/kW presented in Table A1.

In the present decentralized scenario with *Storage CHP*, the power curtailments add up to 3.4%, compared to 33% as calculated for a central gas scenario without any hydrogen storage at all. Decentral electrolysis requires 31% of the total generated power using surpluses from wind (44%), photovoltaics (49%), hydropower (2%), and biomass (5%) (see Figure 4). The surpluses from photovoltaics proportionately predominate since other sources, including wind, are closer to the given demand curve.

The reconversion of hydrogen to power contributes just 12% to the end use in Figure 4. The round-trip efficiency is only 39% instead of the calculated 42% (=70% × 60%). This loss reflects the occasional discrepancies between the (grid) electricity demand and the (local) heat demand mentioned above. Here, the missing heat is supplied by a heating rod from excess electricity or, in the case of a power shortage, through power from hydrogen via the *Storage CHP*. A surplus of heat (*Lost heat*) occurs mostly during electrolysis at summer midday peaks from photovoltaic power. For this reason, higher wind shares have a beneficial effect on the system costs in Figure 3.

In a more advanced version, *Storage CHP* units are directly combined with an electric heat pump on-site, i.e., the heat rod is replaced by a heat pump for increased efficiency in one single device named a *Hybrid CHP*. Since the heat pump's coefficient of performance (COP) is a function of ambient (or water) temperature, combining with the heat from the fuel cell's hydrogen conversion could provide additional benefits that should be the subject of further investigation. In any case, this hybrid configuration of *Storage CHP* and heat pump lowers the overall input power for space heating, and therefore increases the efficiency of the complete system further. The proportion of unused heat is almost halved in Table 1.

**Table 1.** Lost heat and cost reductions. Fuel imports for comparison as reference.

| Scenario | Share of Heaters | Unused Heat | Costs Bn.€/a | Cost Change |
|---|---|---|---|---|
| Fuel import | >95% | 61.7% | 64.9 | Ref. |
| Central Power-to- $H_2$ | 0% | 17.8% | 60.3 | −7.1% |
| *Storage CHP* | 22.4% | 11.5% | 58.9 | −9.2% |
| *Hybrid CHP* | 100% | 6.1% | 56.0 | −13.7% |

In the cost simulation, the hybrid devices reach a share of 100% and reduce the nominal power requirement of photovoltaics, fuel cell, and electrolysis power capacity by 11%, respectively, with the wind again being kept at the 400 GW limit. This reduces the overall system costs by 4.9%.

The average nominal power of the *Hybrid CHP* is 6.3 kW for the fuel cell, 8.6 kW for the electrolysis, and 2.6/4.2 kW for the thermal output, respectively. The calculated hydrogen storage capacity is 4.0 MWh per device (four bundles outdoors, as shown in Figure 1), together with an electric heat pump power of 2.5 kW. Despite the decrease in power and lower costs per device, a *Hybrid CHP* is more complex. From the installer's point of view, the installation is comparable to that of a heat pump, but with twice the footprint of current product designs, and an additional connection to either an outdoor hydrogen bundle (similar to a domestic LPG tank) or to the gas network.

## 3. Energy Transport: Power Grid versus Hydrogen Pipelines

Once network segments or energy cells are operating in equilibrium, as shown in Figure 3, they no longer exchange power with surrounding cells. But what about remote cells producing energy at lower costs due to higher wind shares? What about highly concentrated industrial facilities with highly localized peak loads in a completely decentralized environment? This section compares electricity and gas transport as two equivalent energy transport options.

In a hypothetical case study, southern Germany is modeled as a separate energy cell without wind power, with either with electric or gas connected to the north, while in the previous section, both transport options were assumed to be available in parallel. The sum of the system costs from Section 2 and the additional costs for transport correspond to the levelized cost of electricity (LCOE). The grid fees for electricity and gas are used here as a practical real-world estimate: in Germany, these are currently EUR 0.078/kWh for electricity [24] and EUR 0.016/kWh for gas [25]. The network fees each account for around a third of the total costs for electricity and gas without taxes and levies. Transmission losses are 5.8% for power [26] versus 0.7% for gas in Germany [27,28], which would also favor gas transport, but is not considered here.

With a gradual increase in hydrogen transfer from north to south and separate cost optimizations for both halves, a hydrogen transfer of 700 TWh is required, on which southern Germany can be operated without wind.

This extreme asymmetric system is associated with 25% higher overall costs, including all components in the north and south, but still excluding transport. The share of *Storage CHP* increases from 21% (symmetrical case) to 31% (north) and 84% (south), respectively. The electrolysis capacity increases by a factor of four in the north, but disappears towards zero in the south, i.e., conventional fuel cell CHPs could be used instead. Installed fuel cell powers are nearly symmetric for both the north and south, and curtailments occur only in the north, but decrease overall to 1.9%.

When adding wind-only and PV-only power in Figure 3, there is an increase of 28% in costs or another 3% increase for a complete separation of north and south Germany.

The hydrogen north-to-south-transport of 700 TWh is still below the natural gas consumption of 926 TWh in 2019 [23]. However, due to hydrogen's lower calorific value, higher compressor power would become necessary for larger volumes in cubic meters.

Since the resulting overall system still forms a wind-dominated scenario according to Figure 3, the corresponding hydrogen storage capacity would remain close to 100 TWh, i.e., the current gas network would still be sufficient in terms of volume, but would require increased compressor power.

In contrast to the electricity grid, the peak value for gas transport is not a design nor a cost issue due to the inherent storage capacity within gas networks. Nevertheless, the peak value for gas was 135 GW on 21 January 2019 7 a.m. at an ambient temperature of −2.0 °C. In the present electric scenario with all-electric loads, the north-south transport peaks on 8 December 2018 at 7 a.m. with 161 GW. The total electric energy amounts to 390 TWh/a.

With regard to peak power in GW and annual transported energy in TWh/a, both modes are close to each other. However, since gas transport is five times less expensive and gas pipelines are already in place, the hydrogen transport scenario might outperform the power grid, including expansion, regarding costs. If the figures for transported energy and grid fees are combined, the electric-only scenario is 6% more costly than the gas-only scenario, despite the higher component costs. Furthermore, some wind in the south would be even more cost-effective, especially for south Germany in itself.

This hypothetical case was motivated by much lower wind installations in south Germany currently, together with uncertain future perspectives

Another result from Figure 5 is the basic availability of excess (industrial) hydrogen for steel, chemicals, and heavy transport at a cost of EUR 0.051/kWh or EUR 1.68/kg. An annual industrial demand of 100 TWh would require an additional capacity of 150 GW in photovoltaics and 67 GW in electrolysis without changing wind power capacities.

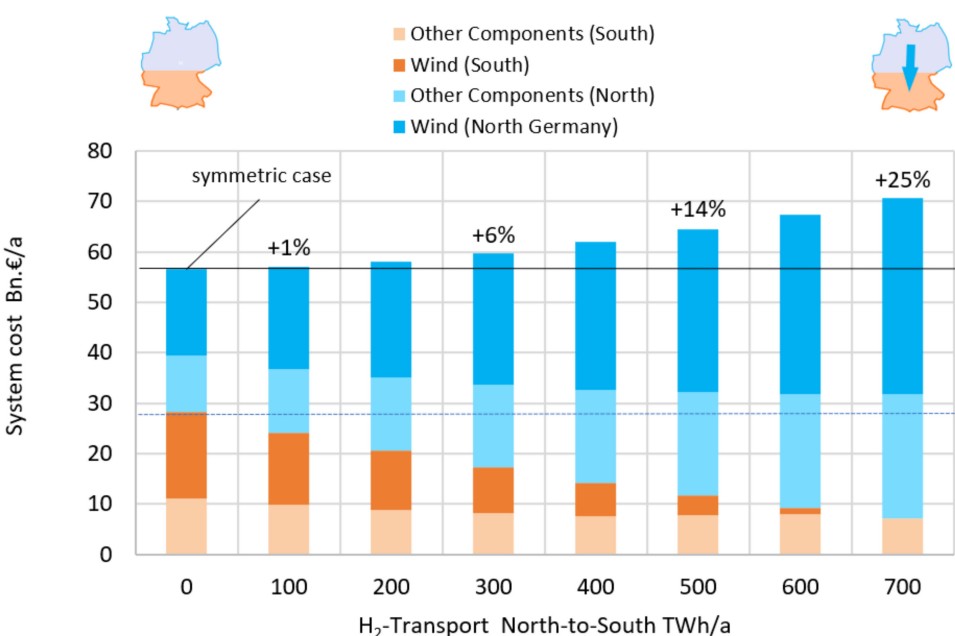

**Figure 5.** Cost structure of an electrically separated Germany versus varied hydrogen transport volumes from north to south.

## 4. Controlled Charging of Electric Cars

The charging profile for electric cars was initially modelled as flat and only slightly temperature-dependent with $-1\%/°C$. In [29], a recurring peak was observed at 6 p.m. for uncontrolled (or wild) charging by private households—see dashed line in Figure 6—with a spread of a factor of 14. In a controlled charging scheme, the charging of electric cars would be limited to 16% of the flat level in times of positive residual load, i.e., whenever renewable electricity is scarce. The 16% follows a more recent estimate [30], according to which this energy proportion is accessed via public charging stations on motorways and in shopping malls as a form of 24-h "must charge" situation. In the case of a negative residual load with a surplus from renewables, a limit of 213% was found to be sufficient. In the simulation, the cars were charging 46% of the time at the lower level, and only 5% of the time did the charging period last longer than 24 h. The two longest low charging periods occurred during a week in November (see Figure 6), and another in December.

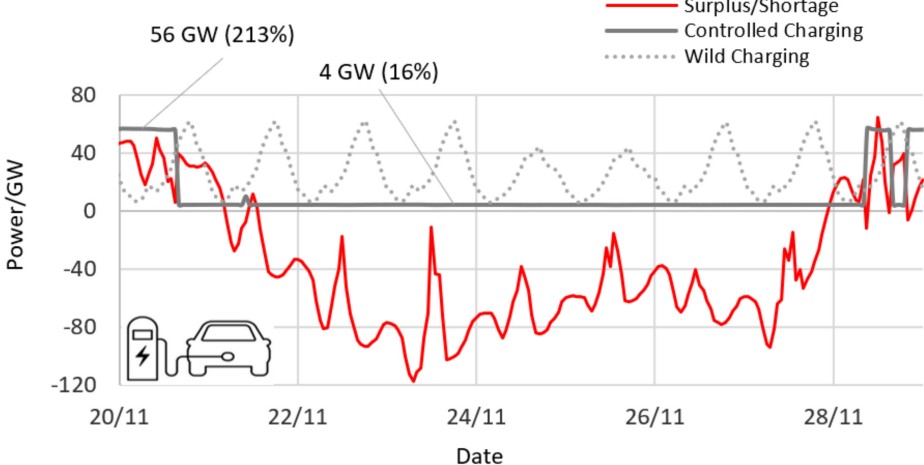

**Figure 6.** Controlled charging (full green) with either 16% or 213%of the average value.

Such gaps would still be feasible with today's ranges of 300 km and average commuting distances of 30 km/day from one full charging period.

Controlled charging would reduce system costs by 5.8%, since less fuel cell power would be required, while wild charging would increase costs by 1.5% over the flat charging system.

At the system level, bi-directional charging/discharging has no effect, since batteries were already not advantageous in terms of system costs, as discussed in Section 2. Alternately, battery costs would now be reimbursed to car owners. Fleet operators can take advantage of more intensive battery usage by charging and discharging when their cars would otherwise not meet the required battery life. In fact, batteries reduce the electricity bills of photovoltaic roof owners. They gain against the grid market price, because this price includes taxes and levies which add up to 40% of the total price in 2021 [24]. However, those private owners will reconnect to the power grid sometime during winter, possibly together with a heat pump. This forms an unpredictable but significant peak to the power grid, and therefore leads increased grid fees.

## 5. Biomass Upgrade to Hydrogen

The last measure to intensify the decentralized hydrogen system is a hypothetical one, since the underlying technology is not yet available on the market.

Up until now, hydropower and biomass were integrated into the model through a flat profile as placeholders for other renewable baseload power plants, e.g., geothermal power. Ideally, such plants should be operated intermittently at times of weak contributions from wind and photovoltaics sources.

The Berlin-based company Graforce [31] advertises an even more attractive option via their low-temperature pyrolysis, called plasmalysis. It electrically cracks hydrocarbons such as methane, plastics, and sewage into hydrogen and carbon. In the case of biomass methane, the process even becomes "$CO_2$ negative". According to the company's website, a microwave process is used with estimated costs equal to electrolysis, but with four times less electrical input for the same amount of hydrogen.

In the model, the methane from biomass was tentatively fed into plasmalysis, and not in a combustion motor. The available biomass feedstock limits the excess power that can be used for plasmalysis instead of electrolysis. For Germany, this is 65 TWh, corresponding to an installed biomass capacity of 20 GW in 2030 (2019: 7.76 GW).

In the standard scenario with 400 GW wind, plasmalysis is therefore limited to 17%, leaving 83% to electrolysis. Of course, this limit can be pushed upwards by using additional residues from forests and agriculture, which might triple the actual biomass availability [32]. According to [31], plastic waste and wastewater could also be used as hydrogen-rich feedstock.

In Figure 7, the system cost decreases by 17.7% for a sixfold increase in available biomass in the 400 GW wind standard scenario. Using plasmalysis within the 20 GW biomass limit leads to a decrease of 11.9% on system costs. For energy cells without wind turbines but with high availability of biomass, the benefit of plasmalysis increases to 24.5%, although the system costs remain much higher for scenarios without wind (see Figure 3).

In summary, plasmalysis is recommendable for energy cells with low wind and high photovoltaic potential when biomass or hydrogen-rich waste is abundant.

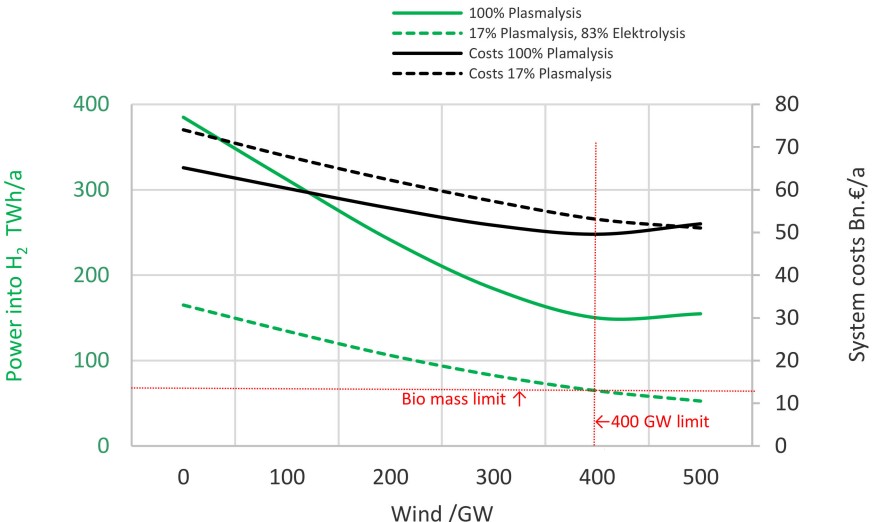

**Figure 7.** Hydrogen from biomass leads to lower system costs. The available biomass sets a limit of 17% for installed plasmalysis power; the rest is left for electrolysis.

## 6. Conclusions and Outlook

The system costs can be reduced by decentral hydrogen storage through effective heat coupling via *Storage CHP* and *Hybrid CHP*. Furthermore, this does not conflict with other flexibility measures, and totals 25.6% in system costs. The single cost reductions in Table 2 seem to be small, since wind turbines and photovoltaics typically account for 70% of the system costs, 19% for *Storage CHP*, 9% for biomass, and 1% for hydropower in the standard 400 GW-wind-scenario.

**Table 2.** Overview of cost reductions across all Sections. The total reduction (25.6%) is slightly larger than the arithmetic sum of all single effects (24.9%).

| Section | Measure | Cost Reduction |
|---|---|---|
| 2 | Central Power-To-Hydrogen | Reference |
| 2 | *Storage CHP* | −2.3% |
| 2 | *Hybrid Storage CHP* with heat pump | −4.9% |
| 4 | Controlled loading | −5.8% |
| 5 | 100% Plasmalysis: −17.7%; 17% Plasmalysis (biomass limit): | −11.9% |
| Σ | **All measures in parallel** | **−25.6%** |

Although hydrogen would be cheaper with power from photovoltaics, wind contributes in a more abundant manner during winter, and this leads to lower overall costs. Furthermore, the potential hydrogen storage capacity remains inside the given storage limits of the current gas network for the standard scenario.

Beyond cost savings, a decentralized hydrogen system enables grid balancing without the need for new infrastructures, since the gas network can be operated with hydrogen as well. It could be demonstrated that power grid expansions become less urgent when hydrogen transport is used instead. The control of the *Storage CHP* is preferably carried out via existing inverter electronics instead of through vulnerable central software. In accordance with the design rules for Industry 4.0, grid services should be operated as autonomously as possible.

The standard scenario developed in this work is in good agreement with the power mix in [5]. In addition to cost advantages, the transition to decentralized hydrogen storage frees the energy system from natural or synthetic gas imports as an integral component in [5]. Synthetic green methane or other green fuels mean lower round-trip efficiencies compared to green hydrogen, no matter who produces them. The systems compatibility argument becomes ambivalent, as 11.3% of the greenhouse effect is caused by the production

and transport of carbonaceous fuels, half of it caused by direct methane emissions from pipelines, but this would not be the case for hydrogen which, as a diatomic molecule, does not contribute to the greenhouse effect at all. A battery storage capacity of 500 GWh was found through a sensitivity analysis only, which is, again, close to [5], but not cost-effective for the standard case as worked out here.

The high battery and photovoltaic contributions found in [5] for a 100% scenario in Europe could not be reproduced under the cost optimization for an isolated Germany. In [6], a well-developed European power grid enables power imports from southern Europe after installing high battery and photovoltaic power capacities installed between Portugal and Turkey. As discussed in Section 3, the electricity grid is not necessarily the most cost-effective solution, and appears to be slow to implement—at least in Germany. Wind in northern Europe, and hydrogen grids in particular, might therefore be the central elements for Europe.

Further work will focus on the prosumer perspective on *Storage* and *Hybrid CHP* as carbon-free heaters to proactively circumvent the problem of heat pump peaks in winter.

**Funding:** This research received no external funding.

**Institutional Review Board Statement:** Not applicable.

**Informed Consent Statement:** Not applicable.

**Data Availability Statement:** Not applicable.

**Conflicts of Interest:** The authors declare no conflict of interest.

## Appendix A

Overall, two conditions must be met for each hour in the year:

$$\overset{Generation}{\underset{i=Wind,\ PV,Hydro,\ Bio}{\sum}} G_i(t) + \overset{Storage}{\underset{i=Battery,H_2}{\sum}} S_i - \overset{Load}{\underset{i=Mobility,Heat,Industry,Power}{\sum}} L_i(t) \geq 0 \qquad (A1)$$

including the hourly generation $G_i(t) = P_{nom,i}[\text{kW}] \cdot \text{energy yield}_i(t) \left[\frac{\text{kWh}}{\text{kW}}\right]$.

The hourly (specific) energy yields are taken from real-world data of ISE Fraunhofer's Energy Charts for Germany in 2018/2019, and scaled up to hourly powers in $G_i(t)$, using the nominal power $P_{nom,i}$ as a free parameter. Boundary conditions are set by total storage capacities for batteries and *Storage CHP,* as defined by their energy-to-power ratios in the footer of Table A1. The electric loads, $L_i$, are scaled up to form an all-electric scenario in 2030, as summarized in Table A2. The contributions of the storage technologies $S_i$ are changing in sign, i.e., they are negative for charging with negative residual loads (i.e., excess power), and positive for discharging with positive residual loads (i.e., power shortages).

Values above zero in (A1) are summed up to the annual curtailments in TWh/a.

The second binding condition for each hour of the year is:

$$a \cdot Q_{Storage\ CHP} + (1-a) \cdot Q_{Heat\ Pumps} - \overset{Load}{\underset{i=Heat,\ Warm\ Water}{\sum}} L_i \geq 0 \qquad (A2)$$

The co-generated heat in the *Storage CHP is* calculated from the efficiency losses in the electrolysis and fuel cells, respectively, i.e., by $(1 - \eta)$. The percentage $a$ of buildings heated by Storage CHP versus $(1 - a)$ by heat pumps is varied as a free parameter. Hourly values above zero are summed up to unused or *lost heat*. Negative values, i.e., operating points with *Missing Heat,* are circumvented by temporarily limiting the efficiency of the *Storage CHP* with electric or hydrogen heating, as explained in Section 2.

The mix of all eight technologies with $P_{\text{nom,I}}$ in kW is summed up using the CAPEX and OPEX data from Table A1 for annual system costs in EUR billion/annum. Excel Solver searches for a minimum in system costs with:

$$S.Cost = \min \sum_{i}^{Techn.} \left( P_{nom,i} \cdot \left( CAPEX_i \cdot \frac{(1+i)^n \cdot i}{(1+i)^n - 1} + OPEX_{i,fix} \right) + OPEX_{i,var} \cdot Op.\,Hours \right) \quad \text{(A3)}$$

using an interest rate of $i = 1.8\%/a$, and where $n$ = device lifetime in years (see Table A3). For the plasmalysis cost estimate in Section 5, the scaled annual cost for power from biomass is used to cover the cost of providing biomethane, along with the cost of the plasmalysis process itself, which has been set as equal to that of electrolysis [31].

**Table A1.** Costs for power generation and conversion for 2030 after [8] pp. 282–285 in "Global Energy System based on 100% Renewable Energy", where the corresponding costs for the fuel cells are adopted from average costs in [12], where it is an increase of 30% versus the electrolysis average.

| Technology (Conversion Efficiency) | CAPEX €/kWel Estim. for 2030 | OPEX fix kW/a | OPEX var €/kWh/a | Lifetime/a |
|---|---|---|---|---|
| PV | 390 | 10.6 | 0 | 35 |
| Wind; onshore | 1000 | 20 | 0 | 25 |
| Elektrolysis * (70%) | 363 | 12.7 | 0.001 | 30 |
| Fuel cell * (60%) | 471.9 | 12.7 | 0.001 | 30 |
| Biomass | 2500 | 175 | 0.001 | 30 |
| Hydro Dam | 1650 | 49,5 | 0.003 | 60 |
| Battery ** (94%) | 150/h | 10/h | 0.0002 | 10 |
| Gas turbine (42%) | 475 | 14.25 | 0.0327 | 30 |

\* Energy-to-power ratio for Storage CHP (Hydrogen): 566–3025 h; ** for batteries: 6 h (in 2030 after [8]).

The heat pumps and two different grades of thermal insulation in the buildings were expressed as a function of the ambient temperature $T_{amb}$ in the reference year 2018/2019. A linear temperature dependence was used for the heat pump's COP (type: Buderus Logatherm WPL 10 Air), while the heat load for a non-renovated and a renovated reference house was modelled by a sigmoidal approximation [33], respectively (see also Figure A1 in [9]).

$$COP_{Heat\ Pump\ WPL\ 10\ A} = 0.0578 \cdot T_{amb} + 2.5245 \quad \text{(A4)}$$

$$Q_{Building\ heat\ demand} = \frac{A}{1 + \left( \frac{B}{T_{amb} - 40} \right)^C} + D \quad \text{(A5)}$$

Non-renovated Building: $A = 243.8$, $B = 37.8$, $C = 8$, $D = 2.92$ Renovated: $A = 96.2$, $B = 38.3$, $C = 8$, $D = 2.92$

$T_{amb}$: data base www.dwd.de/DE/leistungen/klimadatendeutschland/klimadatendeutschland.html#buehneTop (accessed on 7 April 2022).

A renovation progress of 10% was assumed until 2030. The electrical load estimates for transport and industry are taken from [4], and listed in Table A2, including savings.

**Table A2.** Energy consumptions for all sectors in an all-electric system in Germany 2030 before and after thermal insulation and electrification.

| Sector | 2019 TWh | Remarks [4,10,34] | 2030 TWh |
|---|---|---|---|
| Heat | 780 | 900 kWh/person/a with daily warm water profile [34], Temperature: 55 °C→45 °C, Threshold temperature for heating: 13 °C (ambient), Demand reduction by −65% for 10% of the buildings → 497 TWh; Electrification via heat pumps (COP 3.2) →155 TWh | 155 |

**Table A2.** *Cont.*

| Sector | 2019 TWh | Remarks [4,10,34] | 2030 TWh |
|---|---|---|---|
| Mobility | 720 | Electric cars [4] flat profile with −1%/°C; | 212 |
| Industry | 530 | 100% Electrification [4] flat profile | 265 |
| Original Power | 430 | see Figure A1 | 552 |
| **Total** | **2460** | 3289 primary energy 2016 [10]; All-electric sectors → 1184 TWh | **1184** |

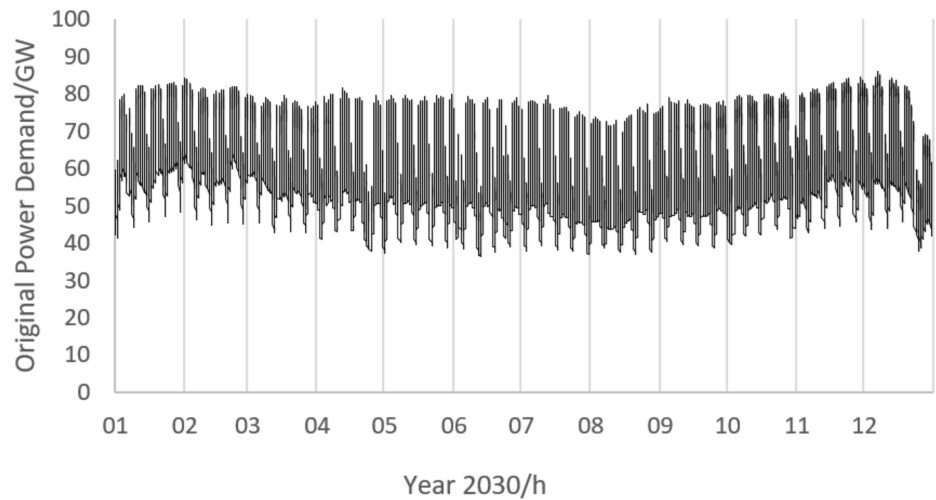

**Figure A1.** Germany's power demand of 552 TWh in 2030 excluding electrification of heat, mobility, and industry, downloaded from www.agora-energiewende.de on 4 July 2016.

**Table A3.** Parameters for generation and *Storage CHP* in the standard scenario with 400 GW wind power. The total cost minimum is found by Excel Solver of Windows 8.1, based on costs from Table A1.

| Technology | Operational Hours h/a | Power/ GW | Costs €/kWh | Costs Bn.€/a |
|---|---|---|---|---|
| PV | 960 | 525 | 0.027 | 13.5 |
| Wind onshore | 2.063 | 400 | 0.034 | 28.0 |
| Elektrolysis | 2.169 | 150 | 0.014 | 6.2 |
| Fuel cell | 1.196 | 203 | 0.029 | 5.1 |
| Biomass | 5644 | 20 | 0.047 | 5.3 |
| Hydro | 5323 | 7 | 0.021 | 0.8 |
| Battery | (838) | 0 | (0.032) | - |
| Gas turbine | (1167) | 0 | (0.063) | - |
| **Total** | | | **0.052** | **58.9** |

The operational hours are based on specific energy yields form ISE Fraunhofer Energy Charts between 6 August 2018 and 5 August 2019 with wind: 123.72 TWh/59.96 GW, photovoltaics: 46.05 TWh/47.96 GW, hydro power: 25.55 TWh/4.80 GW, and power from biomass: 43.80 TWh/7.76 GW, see https://energy-charts.info (accessed on 7 April 2022).

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
