# Peer review of "Decentral Hydrogen"

_energies, doi:10.3390/en15082820_

Round 1

Reviewer 1 Report

First of all, I think the authors provide an interesting paper. This may be a valuable area.

However, I did not find how the authors calculated and obtained these evaluation results in the paper, there is only one formula in the paper, and the full results of the paper cannot be obtained according to it.

As a reviewer, I suggest that the author should give some of the calculation basis in the supplementary material.

In addition, the paper should be said to be quite excellent from content to writing. 

Author Response

Dear Reviewer,

Thank you for reviewing the article "Decentral Hydrogen" energies-1656934:

"As a reviewer, I suggest that the author should give some of the calculation basis in the supplementary material."

The underlying calculations were inserted into the Appendix

Best regards

Paul Grunow

Reviewer 2 Report

Reviewer’s general comment: This article aims to explore the use of hydrogen as energy storage medium and the result of various strategies on total operating cost.

It discusses from the following aspects:
The capacity of renewable energy and energy storage components, the application of advanced version storage CHP, the choice of power and gas transport, the control of battery cars charging, as well as the hypothesis of cracking hydrocarbons into hydrogen and carbon instead of entirely to electronics. 

The scope of this article is very detailed and extensive. In order to help improve the paper quality, my suggestions and comments are shown below.

  • Abstract: good.
  • For the weather condition, energy demand, heat load and some other important data of Germany, the article explains the source of the data, but it is better to show the overall data value in the article.
  • It's necessary to describe the components of the total operation cost through equations or charts in the article to improve readability and leave readers a general understanding.
  • For the energy demand and generation during operation, a time-based energy curve is needed, which can also better illustrate the function of H2 on-site heat coupling in the system.
  • For Figure 3, there should be further explanation to obtain the following heat cover ratio of CHP and heat pump (23% and 77%) as well as the round-trip efficiency of H2 to power (39%). In addition, there is a small error in the figure due to rounding, the energy of storage CHP is unbalanced.
  • The article is mainly discussed based on the description of variable values and results, it is better to give a general control strategy of the system or a general description diagram of the system to improve the readability of the article.
  • In order to improve readability, the hierarchy of the article can be rearranged. For example, the article can begin with identified scientific gaps, and then explain the methodology according to the scientific gaps, finally end with point-by-point contributions of the research.
  • Decentral Hydrogen will be more interesting and meaningful, especially for the high ratio of PV in the electricity market. For example, some applications that can enhance the Decentral Hydrogen are listed below. This can provide justifications on this paper.

Hybrid renewable energy applications in zero-energy buildings and communities integrating battery and hydrogen vehicle storage. Applied Energy 2021.

Transformation towards a carbon-neutral residential community with hydrogen economy and advanced energy management strategies Energy Conversion and Management 2021.

Quantification on fuel cell degradation and techno-economic analysis of a hydrogen-based grid-interactive residential energy sharing network with fuel-cell-powered vehicles. Applied Energy 2021.

Peer-to-peer energy trading of zero-energy communities with hybrid renewable energy systems integrating hydrogen vehicle storage. Applied Energy 2021

Overall, this manuscript is well described. The reviewer suggests the major revision.

Reviewer 3 Report

The idea of this paper "Decentral Hydrogen" is interesting. 
The study extends the power-to-gas approach to small combined heat and power systems in buildings that use fuel cells and electrolysis alternately.
In relation to estimated costs, an optimal mix of photovoltaics, wind, biomass, and hydro-power is calculated.
If you wish to improve the article, you should add a reference and ensure that a proper technical reference list is provided.
Please describe the results in the results section and explain the results obtained by comparing the literature in the discussion section, extending the conclusions and giving practical suggestions.

Author Response

Dear Reviewer,

Thank you for reviewing the articel "Decentral Hydrogen" energies-1656934.

"You should add a reference and ensure that a proper technical reference list is provided."

The technical references [10] to [15] and [28] were added to the reference list.

"Please describe the results in the results section and explain the results obtained by comparing the literature in the discussion section, extending the conclusions and giving practical suggestions."

The results from all Chapters are summarised now in the Chapter "Conclusions and Outlook". The agreements and differences to similar studies, i.e. [4] and [5] in the refernce list, are now disccussed at the end of the same Chapter.

Best regards

Paul Grunow

Round 2

Reviewer 2 Report

Accept